# Discovering Optimal Triplets for Assessing the Uncertainties of Satellite-Derived Evapotranspiration Products

Yan He [1], Chen Wang [1], Jinghao Hu [1], Huihui Mao [1], Zheng Duan [2], Cixiao Qu [1], Runkui Li [1], Mingyu Wang [1] and Xianfeng Song [1,*]

1   College of Resources and Environment, University of Chinese Academy of Sciences, Beijing 100049, China
2   Department of Physical Geography and Ecosystem Science, Lund University, 223 62 Lund, Sweden
*   Correspondence: xfsong@ucas.ac.cn

**Abstract:** Information relating to errors in evapotranspiration (ET) products, including satellite-derived ET products, is critical to their application but often challenging to obtain, with a limited number of flux towers available for the sufficient validation of measurements. Triple collocation (TC) methods can assess the inherent uncertainties of the above ET products using just three independent variables as a triplet input. However, both the severity with which the variables in the triplet violate the assumptions of zero error correlations and the corresponding impact on the error estimation are unknown. This study proposed a cross-correlation analysis approach to discover the optimal triplet of satellite-derived ET products with regard to providing the most reliable error estimation. All possible triple collocation solutions for the same product were first evaluated by the extended triple collocation (ETC), among which the optimum was selected based on the correlation between ETC-based and in-situ-based error metrics, and correspondingly, a statistic experiment based on ranked triplets demonstrated how the optimal triplet was valid for all pixels of the product. Six popular products (MOD16, PML_V2, GLASS, SSEBop, ERA5, and GLEAM) that were produced between 2003 to 2018 and which cover China's mainland were chosen for the experiment, in which the error estimates were compared with measurements from 23 in-situ flux towers. The findings suggest that (1) there exists an optimal triplet in which a product as an input of TC with other collocating inputs together violate TC assumptions the least; (2) the error characteristics of the six ET products varied significantly across China, with GLASS performing the best (median error: 0.1 mm/day), followed by GLEAM, ERA5, and MOD16 (median errors below 0.2 mm/day), while PML_V2 and SSEBop had slightly higher median errors (0.24 mm/day and 0.27 mm/day, respectively); and (3) removing seasonal variations in ET signals has a substantial impact on enhancing the accuracy of error estimations.

**Keywords:** satellite-derived evapotranspiration products; error assessment; extended triple collocation; cross-correlation analysis; optimal triplet



## 1. Introduction

Evapotranspiration (ET) is a key component of the water and energy balance in climate–soil–vegetation interactions, controlling 60–65% of water loss from rainfall inputs [1], and up to 90% of water loss in arid regions [2]. Advanced remote sensing (RS) sensors such as MODIS and VIIRS [3] are effective in characterizing various surface fields and land surface models, and data assimilation techniques are also improving. Nevertheless, large differences are observed between satellite-derived or reanalyzed ET products, and it is unclear why these differences arise and how they affect applications such as water resources, agricultural systems, and ecosystem management, which are heavily dependent on hydrometeorological conditions [4]. Therefore, assessing errors in various ET products is critical to understanding the impact of their uncertainties on these applications.

Currently, two main error estimation techniques are used to assess the uncertainty of geophysical variables [5] and can be used to estimate errors in ET products. The first method is mainly used for verification, in which the error is characterized by calculating the correlation, root mean square error, and other indicators between ET products and in-situ measurements. However, obtaining ET observations from in-situ flux towers is challenging due to the lack of sufficient instrumented sites in most regions. The second approach is the triple collocation (TC) error estimation technique [6], which uses statistical relationships to estimate the standard deviation (STD) of the random errors of three or more identical geophysical datasets as variables in the absence of truth values; this method is often used as a powerful supplementary validation scheme for those regions with limited in-situ sites when the input datasets conform to the TC assumptions.

TC methods have become very popular in recent years and are still being developed. It was originally designed by Stoffelen [6] for near-surface wind velocities, and then McColl, et al. [7] developed the extended triple collocation (ETC) method, which introduced a new variable to avoid setting one of the inputs as the reference in solving the TC equation, thus providing more reliable indicators such as the correlation coefficient, the signal-to-noise ratio, and the fRMSE metric [8]. Generalizing the triple collocation analysis to an arbitrary number of input datasets has also been explored, but it is still subject to the general assumption of independent errors in TC. Pierdicca, et al. [9] proposed an extended quadruple collocation (E-QC) approach that integrates quadruple collocation (QC) [10] and extended collocation (EC) [11] to process four more inputs, aiming at relaxing the hypothesis of statistically independent errors. Nevertheless, it still needs to know one input in advance to be independent from the others as a priori and to ensure that each member of the dataset, which exhibits a non-zero error cross-correlation, is also included in at least one dataset triplet where the errors are fully independent [9]. The most practical procedure that extends the triple collocation to multiple collocation (MC) by Pan, et al. [12] can handle any number of data sources under the framework of Pythagorean constraints in the Hilbert space, which is a complete inner product space capable of defining geometric concepts such as length, angle, and orthogonality. The error assessment for an input source is equivalent to the mean of all possible TC solutions under Pythagorean constraints that one can perform against the other N-1 inputs. It does output a unique solution to the multiple collocation but it by no means resolves the potentially discrepant conclusions from individual triple collocations. It remains unknown as to whether it actually provides any better-performed error estimation since this extension still cannot answer how much the uncorrelated error assumption is being violated [12].

Currently, the TC analysis has been widely used in various geographic variables, including soil moisture products [11,13,14], precipitation [15–18], the leaf-area index [19], land-based water storage [20], marine gravity [21], and ET [5]. In terms of the number of publications, compared with soil moisture and precipitation products in the water cycle, relatively few papers consider the application of the TC in the estimation of errors in ET or vapor flux products [22]. ET products display spatial variations in errors that are significantly influenced by the characteristics of land use and land cover (LULC). Research conducted by Khan, Liaqat, Baik, and Choi [5] has shown that the MOD16 estimations exhibit good agreement with ground observations in rice paddy ecosystems. In contrast, the GLEAM product exhibits lower bias errors in forest and grassland biomes. Currently, the commonly utilized evapotranspiration datasets can be broadly classified into the following three categories [23]: (1) fully physically-based combination models that incorporate principles of mass and energy conservation; (2) semi-physically based models that focus on either mass or energy conservation; and (3) black-box models that rely on artificial neural networks, empirical relationships, as well as fuzzy and genetic algorithms. Unlike precipitation and soil moisture products, ET products are generated based on multiple meteorological forcing and land surface datasets. These ancillary datasets can be shared by many products [24–29], violating the basic assumption of zero-error correlations

to varying degrees. Violating the assumptions about the structure of input datasets affects the stability and reliability of the estimation of random errors, which in turn limits the efficacy of the multiple collocation analysis. However, the extent of the violations of the non-zero cross-correlation hypothesis and their implications for the accuracy of the multiple collocation analysis remain uncertain. None of the existing TC extensions can completely overcome this problem due to the weakness of the fundamental assumption of the TC [9,12].

In addition to error assessment, the TC method is valuable for data fusion, where original datasets are combined to generate more accurate data products using the TC-estimated errors. The integration of the least square method with the TC method has proven effective for fusing diverse data products [12,20]. This approach enhances the accuracy of datasets without relying on user-defined parameters and has been widely applied in various fields, including soil moisture [14,30,31], evapotranspiration [5,32], land storage anomaly [20], precipitation [33], and so on.

To investigate and generate a more reliable and accurate ET estimate, the satellite-based or reanalyzed ET products are fused in this work, based on error characteristics calculated by the ETC technique [3,15]. ETC is deployed because it can provide the acceptable error estimation by discovering the optimal triplets, plus more quality indicators, provided that the in-situ measurement is available in our experiment. Other multiple collocation analyses are not appropriate for our situation. The general extension of MC by Pan, et al. is a kind of "voting" process and one input data source gets one vote [12], under which the error estimation is an averaged solution among all possible triplets, not the optimum, which is practical in mathematics for a general solution without the support of in-situ datasets. The E-QC and QC request one input to be independent from others and also take it as a unit of reference; thus, setting an ET dataset as a reference will result in a biased estimation. Moreover, E-QC needs at least three inputs with independent errors as the EC required to keep a number of zero correlated dataset pairs, which is also a big challenge for ET products to simultaneously obtain three wholly independent datasets due to the errors of ET products often being inevitably cross-correlated.

In this paper, we first introduce six popular ET products and in-situ flux datasets, then describe the ETC method and the approach to determine the optimal triplet among all possible triplets in which the dataset pairs violate the collocation assumptions to varying degrees; we ranked the triplets based on the correlation between ETC-based and in-situ-based error metrics, and, finally, determined the best error estimate for a product, which was calculated at the product's original spatial resolution and validated by the statistics related to the differences of the ranked error estimates.

## 2. Study Area and Datasets

### 2.1. Study Area

The research area is China's mainland ($73°33'$–$135°05'$E, $3°51'$–$53°33'$N), about 5200 km from east to west and 5500 km from north to south, with a stepped distribution of high elevation in the west and low elevation in the east (Figure 1). The climate is complex and diverse; precipitation mostly occurs in summer. The ET is the evaporation of water trapped in water bodies, soils, and vegetation surfaces, as well as transpiration by plants. Therefore, the ET varies regionally due to the significant differences in climate, vegetation, and soil across the study area [34], leading to strong spatial and temporal heterogeneity in different regions at different times.

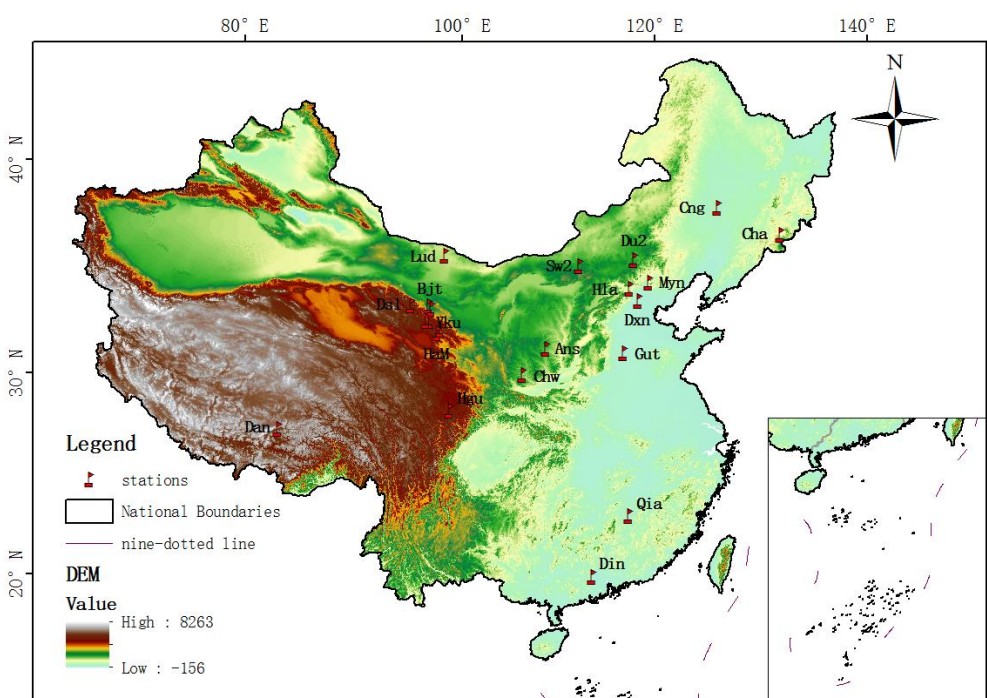

**Figure 1.** Study area and in-situ flux measurement sites.

*2.2. Datasets*

2.2.1. Evapotranspiration Datasets

Six satellite-derived or reanalyzed ET products were collected to assess their quality, as shown in Table 1. These datasets have varying degrees of homology often attributable to a common origin, in terms of the model principles of ET inversion or the model inputs, resulting in errors in these datasets that cannot be completely independent of each other. In addition, the spatial scales of the above ET products are also different. The specific information is as follows.

**Table 1.** Characteristics of the evapotranspiration datasets.

| Datasets | Scheme/Model | Spatial Resolution | Temporal Resolution | Date Range |
|---|---|---|---|---|
| MOD16A2GF | Penman–Monteith | 500 m | Every 8 days | Jan.2000–Dec.2021 |
| PML_V2 | Penman–Monteith–Leuning | 500 m | Daily | Feb.2000–Dec.2020 |
| GLASS | Bayesian model averaging | 1 km | Every 8 days | Jan.2000–Dec.2018 |
| SSEBop | surface energy balance | 1 km | Every 10 days | Jan.2003–Jun.2021 |
| ERA5 | IFS | 0.1° | Hourly | Jan.2001–present |
| GLEAM | Priestley Taylor | 0.25° | Daily | Jan.2003–Jul.2020 |

MOD16A2GF v061 [24] (www.usgs.gov, accessed on 18 June 2023) calculates the ET according to Penman–Monteith using daily meteorological reanalysis data and the eight-day remote sensing vegetation dynamics of MODIS. However, it has been observed that MOD16 exhibits significant deviations from local-scale observations [35,36]. Therefore, in this study, we utilize the gap-filled version of MOD16A2, where cloud-contaminated LAI/FPAR gaps are temporally filled before calculating the ET. However, it is important to note that this version still has a substantial bias, as illustrated in Figure 2.

PML_V2 [25] (data.tpdc.ac.cn, accessed on 18 June 2023) adopts GLDAS 2.1 meteorological data and MODIS MCD12Q2.006 IGBP reflectance, emissivity, LAI, and continuous dynamic vegetation as inputs for a Penman–Monteith–Leuning model for estimating terrestrial ET and the total primary productivity dataset.

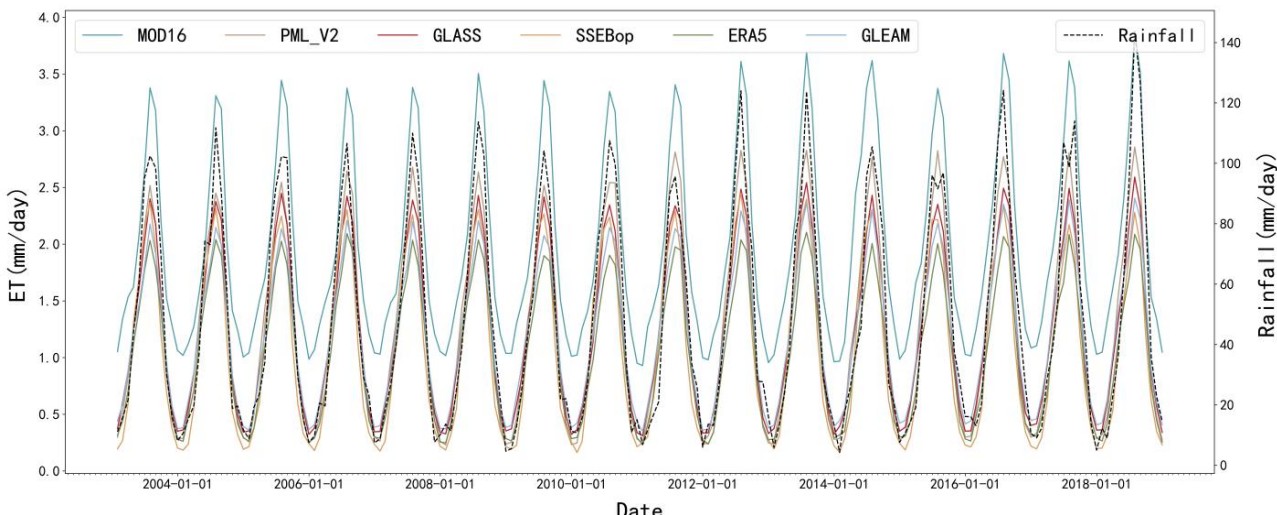

**Figure 2.** The time series of the rainfall in China with the variation of evapotranspiration.

GLASS [26] (www.GLASS.umd.edu, accessed on 18 June 2023) uses a Bayesian model-averaging (BMA) approach that incorporates five latent heat flux algorithms to estimate the ground latent heat flux (LE), including MOD16, RRS-PM, PT-JPL, MS-PT, and UMD-SEMI.

SSEBop [27] (edcintl.cr.usgs.gov, accessed on 18 June 2023) applies a simplified surface energy balance (SSEB) model with predefined "hot" and "cold" boundary conditions to estimate ET per pixel. MODIS relative data and GLDAS meteorological data are used as the model inputs.

ERA5-Land [28] (cds.climate.copernicus.eu, accessed on 18 June 2023) is a reanalysis dataset that uses variables, such as temperature, generated by the land component of the ECMWF ERA5 climate as a meteorological driver and calculates the potential ET based on the surface energy balance.

GLEAM v3.5b [29] (www.gleam.eu, accessed on 18 June 2023) is a complex surface-model product that first uses the Priestley–Taylor equation to calculate potential evaporation from surface net radiation and near-surface air temperature, and then obtains actual evaporation data from the multiplicative evaporative stress factor. It uses CERES, TMPA, and AIRS radiation, rainfall, and temperature data as model inputs to estimate the different components of land evaporation.

### 2.2.2. In-Situ Datasets

The in-situ data in this study were obtained from multiple sources, including the FLUXNET2015 dataset [37] (https://fluxnet.org, accessed on 18 June 2023), the multi-scale surface flux and meteorological elements observation dataset in the Hai River basin [38] (http://data.tpdc.ac.cn, accessed on 18 June 2023), and the CERN original observation dataset (http://rs.cern.ac.cn, accessed on 18 June 2023). Based on the data availability of in-situ measurements, a total of 23 stations across the Chinese mainland (Figure 1) are selected, and the details of the station information are shown in Table 2.

**Table 2.** The in-situ flux tower sites.

| | Site | Latitude | Longitude | Type | Temporal Extent | Province | Data Source |
|---|---|---|---|---|---|---|---|
| 1 | Cha | 42.40 | 128.10 | Forest | 2003–2005 | Jilin | Fluxnet |
| 2 | Din | 23.17 | 112.54 | Forest | 2003–2005 | Guangdong | Fluxnet |
| 3 | Qia | 26.74 | 115.06 | Forest | 2003–2005 | Fujian | Fluxnet |
| 4 | Cng | 44.59 | 123.51 | Grassland | 2007–2010 | Jilin | Fluxnet |
| 5 | Dan | 30.50 | 91.07 | Grassland | 2004–2005 | Xizang | Fluxnet |
| 6 | HaM | 37.37 | 101.18 | Grassland | 2003–2004 | Qinghai | Fluxnet |

**Table 2.** *Cont.*

|    | Site | Latitude | Longitude | Type | Temporal Extent | Province | Data Source |
|----|------|----------|-----------|------|-----------------|----------|-------------|
| 7  | Hgu  | 32.85    | 102.59    | Grassland | 2015–2017  | Sichuan   | Fluxnet |
| 8  | Du2  | 42.05    | 116.28    | Grassland | 2006–2008  | Neimenggu | Fluxnet |
| 9  | Du3  | 42.06    | 116.28    | Grassland | 2009–2010  | Neimenggu | Fluxnet |
| 10 | Sw2  | 41.79    | 111.90    | Grassland | 2010–2012  | Neimenggu | Fluxnet |
| 11 | Aru  | 38.05    | 100.46    | Grassland | 2014–2017  | Qinghai   | TPDC |
| 12 | Dsl  | 38.84    | 98.946    | Grassland | 2014–2017  | Qinghai   | TPDC |
| 13 | Yku  | 38.01    | 100.24    | Grassland | 2015–2017  | Qinghai   | TPDC |
| 14 | Hla  | 40.35    | 115.79    | Cropland  | 2014–2017  | Hebei     | TPDC |
| 15 | Myn  | 40.63    | 117.32    | Cropland  | 2008–2010  | Beijing   | TPDC |
| 16 | Gut  | 36.52    | 115.13    | Cropland  | 2008–2010  | Hebei     | TPDC |
| 17 | Dxn  | 39.62    | 116.43    | Cropland  | 2008–2010  | Beijing   | TPDC |
| 18 | Chw  | 35.24    | 107.68    | Cropland  | 2010–2015  | Shanxi    | CERN |
| 19 | Ans  | 36.86    | 109.32    | Cropland  | 2016–2017  | Shanxi    | CERN |
| 20 | Ha2  | 37.61    | 101.33    | Wetland   | 2003–2005  | Qinghai   | Fluxnet |
| 21 | Ssw  | 38.79    | 100.49    | Desert    | 2013–2014  | Gansu     | TPDC |
| 22 | Bjt  | 38.91    | 100.30    | Desert    | 2014–2014  | Gansu     | TPDC |
| 23 | Lud  | 42.00    | 101.13    | Desert    | 2014–2015  | Neimenggu | TPDC |

To match satellite-based or reanalyzed ET datasets, in-situ ET data were aggregated on an eight-day scale. The preprocessing of half-hourly interval observations from Fluxnet [37] was carried out to select those data with good measurements and filling conditions, thus controlling the quality of the flux tower data and excluding the day of rainfall and the day after to avoid canopy interception evaporation and sensor saturation effects. The unit of latent heat flux was converted to mm/day. The measured ET data from TPDC were obtained using the eddy covariance (EC) system and large-aperture scintillator (LAS) observations. Due to the existence of measurement errors, the quality control on the selection of eddy data followed the processing procedures [38]. In addition, large-scale lysimeter observation data (mm/day) were directly obtained from the resource-sharing service platform of China CERN.

## 3. Methodology

### 3.1. Extended Triple Collocation

Triple Collocation (TC) is a statistical technique that analyzes errors within a triplet of spatially coincident or temporally collocated datasets of the same geophysical variable, without knowledge of the truth values. It provides insights into individual error statistics and underlying uncertainties. The TC method is based on four assumptions related to its input signals: (i) linearity between the modeled signals and the truth signals, (ii) the stationary nature of the modeled signal and error statistics, (iii) the errors among the modeled signals are independent and uncorrelated, and independent on truth signals, and (iv) error orthogonality, that is, the random error in the modeled signals is independent of the true values [7]. The model is listed below:

$$X_i = X_i' + \varepsilon_i = \alpha_i + \beta_i t + \varepsilon_i \tag{1}$$

where $X_i (i \in \{1, 2, 3\})$ are three collocated independent ET inputs, $t$ is the true ET signal but is unknown, $\alpha_i$ and $\beta_i$ are the systematic biases relative to the true signal, $\alpha_i$ is the additive bias, $\beta_i$ is the multiplicative bias, and $\varepsilon_i$ is the zero-mean random error of the modeled signals.

Based on Equation (1), the covariance between different ET systems can be expressed as:

$$Cov(X_i, X_j) = E(X_i X_j) - E(X_i)E(X_j) = \beta_i \beta_j \sigma_t^2 + \beta_i Cov(t, \varepsilon_j) + \beta_j Cov(t, \varepsilon_i) + Cov(\varepsilon_i, \varepsilon_j) \tag{2}$$

where $\sigma_t^2 = Var(t)$. Assuming that the errors of different ET datasets are not correlated with each other or with the truth values, it yields zero covariances among the errors and truth values $(Cov(\varepsilon_i, \varepsilon_j) = 0, i \neq j)(Cov(\varepsilon_i, t) = 0)$. As a result, the last three terms of Equation (2) are eliminated, and thus are simplified to:

$$Q_{ij} \equiv Cov(X_i, X_j) = \begin{cases} \beta_i \beta_j \sigma_t^2, & \text{for } i \neq j \\ \beta_i^2 \sigma_t^2 + \sigma_{\varepsilon_i}^2, & \text{for } i = j \end{cases} \tag{3}$$

where $\sigma_{\varepsilon_i}^2 = Var(\varepsilon_i)$. The above covariance matrix contains six unique terms $(Q_{11}, Q_{12}, Q_{13}, Q_{22}, Q_{23}, Q_{33})$, but there are seven unknowns $(\beta_1, \beta_2, \beta_3, \sigma_{\varepsilon_1}, \sigma_{\varepsilon_2}, \sigma_{\varepsilon_3}, \sigma_t)$; therefore, there is no solution for Equation (3). The traditional TC method needs to set one of the ET datasets as the reference, and then convert the other datasets to the reference space, that is, $\alpha_1 = 0$ and $\beta_1 = 1$, which simplifies Equation (3) and makes it solvable.

However, this simple rescaling of the ET system with simplified estimation equations leads to biased estimates of RMSE [6]. For this reason, McColl, Vogelzang, Konings, Entekhabi, Piles, and Stoffelen [7] proposed the extended triple collocation method, which reduces the number of unknowns and avoids the simple scaling process seen in the traditional TC by defining a new variable, $\theta_i = \beta_i \sigma_t$. The creative simplification of Equation (4) is as follows:

$$Q_{ij} = \begin{cases} \theta_i \theta_j, & \text{for } i \neq j \\ \theta_i^2 + \sigma_{\varepsilon_i}^2, & \text{for } i = j \end{cases} \tag{4}$$

The standard deviation (STD) of the random variable as the essential output of the TC analysis can thus be solved as shown below (Equation (5)):

$$\sigma_\varepsilon = \begin{bmatrix} \sqrt{Q_{11} - \frac{Q_{12}Q_{13}}{Q_{23}}} \\ \sqrt{Q_{22} - \frac{Q_{12}Q_{23}}{Q_{13}}} \\ \sqrt{Q_{33} - \frac{Q_{13}Q_{23}}{Q_{12}}} \end{bmatrix} \tag{5}$$

More importantly, $\theta_i$ can be used to solve the correlation coefficient of the modeled ET variable concerning the unknown ET truth values. Both $\theta_i$ (Equation (6)) and the correlation coefficient $\rho_{t,X}$ (Equation (7)) can be obtained through the ordinary least square method, and the signal-to-noise ratio can be calculated using the square of $\rho_{t,X}$ (Equation (8)):

$$\theta_i = \rho_{t,X_i} \sqrt{Q_{ii}} \tag{6}$$

$$\rho_{t,X} = \pm \begin{bmatrix} \sqrt{\frac{Q_{12}Q_{13}}{Q_{11}Q_{23}}} \\ sign(Q_{13}Q_{23})\sqrt{\frac{Q_{12}Q_{23}}{Q_{22}Q_{13}}} \\ sign(Q_{12}Q_{23})\sqrt{\frac{Q_{13}Q_{23}}{Q_{33}Q_{12}}} \end{bmatrix} \tag{7}$$

$$\rho_{t,x_i}^2 = \frac{\beta_i^2 \sigma_t^2}{\beta_i^2 \sigma_t^2 + \sigma_{\varepsilon_i}^2} = \frac{SNR_{ub}}{SNR_{ub} + 1} \tag{8}$$

*3.2. Experimental Design*

3.2.1. Data Standardization

TC methods are dedicated to evaluating the standard deviation (STD) of the random errors of a dataset, and they cannot produce any information about the systematic error (bias) or the presence of time-varying biases caused by seasonal variations in the dataset, which affects the accuracy of assessing random errors using the TC analysis [39]. Because both the signal and the error of the ET datasets contain strongly seasonally variation information as is shown partly in Figure 2, which seriously violates the stability assumption of the TC, the bias caused by seasonal changes must be eliminated during the data prepro-

cessing. Otherwise, non-linear and non-zero error cross-correlation could occur between the input ET products within a triplet [40].

The data can be decomposed into the mean and anomaly components, as shown in Equation (9); the anomaly data, relative to the seasonal signal, are used as the input data, for which the systematic deviation is included in the mean value [41]. In this way, we can effectively alleviate the volatility of the ET datasets and avoid violating the TC assumption of signal stationarity [11]. Most studies posit that decomposition using the monthly scale as a moving window can effectively remove the seasonal information of time-series data [42].

$$\theta_X = <\theta_X>_D^N + \theta_x' \tag{9}$$

where $<\theta_X>_D^N$ is the mean value of the ET signal; the mean value at each moment is calculated by the time window N centered on this moment; and $\theta_x'$ is the anomaly term corresponding to the mean value. The time scale of the ET dataset is 8 days, so N is set to 4.

### 3.2.2. Cross-Correlation Analysis

Considering the six ET products shown in Table 1, there are a number of possible triplets that can be used to estimate the errors of a product using the ETC analysis. This is because these products share meteorological forcing inputs or land surface datasets, and their errors will inevitably be correlated. Thus, the error estimated by ETC for the same product may differ depending on which other two products are its counterparts. In this situation, it is necessary to consider which combination of triplets can result in the best estimation of errors for a given product in comparison of the observation at in-situ sites, and how to determine the triplet that can obtain the most reliable spatial distribution of the product error. These concerns constate the focus of the experimental design.

Most studies carry out the TC analysis on spatially aggerated datasets if the spatial resolutions of the products differ from each other. The impact of the representative error for a product on the TC analysis may be larger than the one caused by the non-zero error correlations between the product and its two counterparts if the product is spatially aggregated from a very fine resolution to a quite coarse resolution. So, as shown in Table 3, this study constructed the triplets for an ET product as possible at its original spatial resolution to reduce the impact of scale conversion on the accuracy of the product due to the neutralization of neighbor cell values. Meanwhile, the products that have temporal resolutions different from that of MODIS-ET were also generated uniformly to the eight-day interval using temporal aggregation. In addition, there are only two 500 m datasets (MOD16 and PML_V2) available, and we have upsampled two 1 km datasets (GLASS and SSESop) to perform the ETC analysis at the finest 500 m resolution, providing that the conversion between such close resolutions causes tiny errors on spatial representativeness.

**Table 3.** The correlation coefficients between the ETC-estimated error and the actual error.

| | Triplets | MOD16 500 m | PML_V2 500 m | GLASS 1 km | SSEBop 1 km | ERA5 0.1° | GLEAM 0.25° |
|---|---|---|---|---|---|---|---|
| 1 | MOD16-PML_V2-GLASS | 0.448 (0.23) | 0.871 (0.24) | 0.442 (0.11) | | | |
| 2 | MOD16-PML_V2-GLEAM | | | | | | 0.551 (0.15) |
| 3 | MOD16-PML_V2-SSEBop | 0.725 (0.22) | 0.874 (0.24) | | 0.799 (0.27) | | |
| 4 | MOD16-PML_V2-ERA5 | | | | | 0.062 (0.14) | |
| 5 | MOD16-GLASS-GLEAM | | | | | | 0.893 (0.18) |
| 6 | MOD16-GLASS-SSEBop | 0.821 (0.19) | | 0.472 (0.13) | 0.856 (0.27) | | |
| 7 | MOD16-GLASS-ERA5 | | | | | 0.880 (0.19) | |
| 8 | MOD16-ERA5-GLEAM | | | | | | 0.525 (0.13) |
| 9 | MOD16-SSEBop-GLEAM | | | | | | 0.890 (0.17) |
| 10 | MOD16-SSEBop-ERA5 | | | | | 0.850 (0.19) | |
| 11 | PML_V2-GLASS-GLEAM | | | | | | 0.380 (0.12) |
| 12 | PML_V2-GLASS-SSEBop | | 0.829 (0.20) | 0.668 (0.10) | 0.854 (0.26) | | |

**Table 3.** *Cont.*

| Triplets | | MOD16 500 m | PML_V2 500 m | GLASS 1 km | SSEBop 1 km | ERA5 0.1° | GLEAM 0.25° |
|---|---|---|---|---|---|---|---|
| 13 | PML_V2-GLASS-ERA5 | | | | | 0.204 (0.12) | |
| 14 | PML_V2-SSEBop-GLEAM | | | | | | 0.660 (0.13) |
| 15 | PML_V2-ERA5-GLEAM | | | | | | 0.502 (0.11) |
| 16 | PML_V2-SSEBop-ERA5 | | | | | 0.000 (0.14) | |
| 17 | GLASS-SSEBop-GLEAM | | | | | | 0.691 (0.14) |
| 18 | GLASS-ERA5-GLEAM | | | | | | 0.730 (0.11) |
| 19 | GLASS-SSEBop-ERA5 | | | | | 0.759 (0.17) | |
| 20 | SSEBop-ERA5-GLEAM | | | | | | 0.486 (0.12) |

The values in the table denote the correlation coefficients (the median of ETC-estimated STD errors of all pixels).

The ETC analysis was performed on all triplets, enabling the estimation of the standard deviation of the random error (referred to as ETC-estimated STD errors) for each product at its original spatial resolution. Assuming that the ET data measured at in-situ towers can be regarded as the true values, the actual errors of each satellite-derived ET product at all tower locations were calculated, and the standard deviation of these actual errors, tentatively referred to as actual STD errors, was estimated. To assess the performance of each triplet in the TC analysis, we performed a linear correlation analysis with ETC-estimated and actual STD errors for each ET product at its original spatial resolution. By ranking the triplets with the correlation coefficient as the score, the triplet with the highest coefficient was treated as the optimum, as it produced the best error estimation. The flow chart is shown in Figure 3.

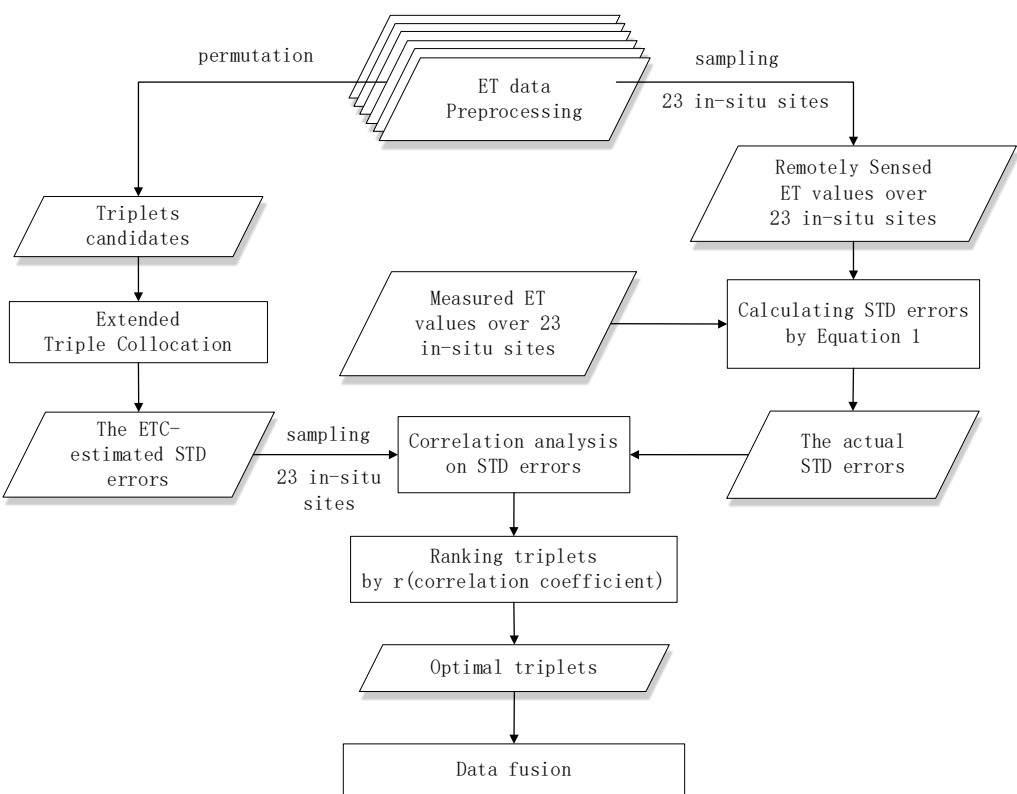

**Figure 3.** Flow Chart.

### 3.2.3. Evaluation Indicators

Previous studies have compared the difference between the TC-estimated random error and the RMSE that is calculated between the satellite-derived ET product and the measured flux data, in order to verify whether the ETC error estimation is robust. They are

not comparable if it is necessary to review Equation (6) in the TC method [43]. Indeed, the comparison focuses on the standard deviation of the product random errors.

(1) The correlation coefficient $r$ (Equation (10)) is the amount of linear correlation between two variables: the larger the $r$, the higher the degree of correlation between two variables.

$$r = \frac{\sum\limits_{i=1}^{N} \left(S_i - \overline{S}\right)^2 \left(M_i - \overline{M}\right)^2}{\sqrt{\sum\limits_{i=1}^{N} \left(S_i - S\right)^2} \sqrt{\sum\limits_{i=1}^{N} \left(M_i - M\right)^2}} \tag{10}$$

(2) The root mean square error (RMSE) is used to test the difference between two variables. The smaller the RMSE (Equation (11)), the smaller the difference between the two variables [5].

$$\text{RMSE} = \sqrt{\frac{1}{n}\sum\limits_{i=1}^{n}(M_i - S_i)^2} \tag{11}$$

where $S_i$ is the estimated value, $M_i$ is the actual value, and $n$ stands for the number of in-situ sites.

### 3.2.4. Data Fusion

A data fusion method is proposed to merge the satellite-derived ET products by following a popular fuzzy membership merging method [44]. Under a fuzzy assignment, the fuzzy membership stands for the similarity between the source data and their counterpart target. The more similar a source is to the target, the stronger the assignment. An uncertainty-reduced ET estimate can be made for the original ET products using either a hardening (Equation (12)) or weighing (Equation (13)) mode, as follows:

$$ET_{harden} = \max(ET_i) \tag{12}$$

$$ET_{weigh} = \sum\limits_{i=1}^{N} w_i ET_i \tag{13}$$

where $w_i$ is the fuzzy membership of the dataset $ET_i$, $\sum\limits_{i=1}^{N} w_i = 1$, and $N$ is the number of members.

The calculation of the fuzzy membership is based on an objective-based least-square method [31], which has been successfully used to merge satellite-derived and model-based soil moisture products. The basis of the least-square method is to minimize the summary of the error variance of all the merged estimates by the expression of $w_i$ (Equation (14)) [45]:

$$w_i = \frac{\left(\sigma_i^{-1}\right)^2}{\sum\limits_{i=1}^{N} \left(\sigma_i^{-1}\right)^2} \tag{14}$$

where $\sigma_i$ is the standard deviation of the random error of the $i_{th}$ ET product estimated by the TC analysis.

## 4. Results

### 4.1. Optimal Triplet Mining by Correlation Analysis

For all the triplets constructed for an ET product in the experiment, we first used the ETC method to estimate the STD of random errors at the original resolution of the product, and then compared them with the actual STDs at 23 in-situ flux sites using the correlation analysis [39]. By ranking the performance of the triplets in the product with the correlation coefficient, the optimal triplet for the product at a given spatial resolu-

tion was identified to estimate the product's errors (Table 3). Taking MODIS-ET as an example, the best triplet occurs with the combination of MOD16–GLASS–SSEBop, which leads to a top correlation coefficient of 0.82 and a median of 0.19 for ETC-estimated STD errors. The magnitude of the correlation coefficient implies whether the ETC-estimated errors are consistent with the actual errors recorded in-situ. Interestingly, the products of MOD16, GLASS, ERA5, and GLEAM show large differences in their correlation coefficients, indicating that they could seriously be affected by their different counterparts in the TC analysis. In comparison, PML_V2 and SSEBop are much more reliable for the TC analysis according to the correlation coefficient. Moreover, aggregating high-spatial-resolution products to match low-resolution datasets reduces the accuracy of ETC estimations [22], but some high collocating triplets are still found in this work, i.e., ERA5 and GLEAM.

### 4.2. Spatial Distribution of the ETC-Estimated STD Errors of Six ET Products

Each of the six satellite-derived ET products has one optimum triplet, which was identified for the spatial resolution of the product. By evaluating the error estimates that result from these triplets using ANOVA (Table 4), the *p*-values of the F-statistics show that the correlation between the ETC-estimated and actual STD errors was significantly high at the significance level of 0.05; meanwhile, the RMSE is quite low. The 500 m products of MOD16 and PML_V2 were collocated in the TC analysis with the upsampling versions of the 1 km GLASS and SSEBop; it seems that the impact of the scale shifting on the TC analysis is lower than that of the impact caused by violations of the TC assumptions. In addition, it is worth noting that the slopes of the best combination are all less than one for all products, suggesting that the ETC method as a whole may underestimate the product error; this finding is similar to that regarding the TC analysis of soil moisture products [46].

**Table 4.** ANOVA analysis of the correlation between the ETC-estimated and actual STD errors for six products.

| | MOD16 | PML_V2 | GLASS | SSEBop | ERA5 | GLEAM |
|---|---|---|---|---|---|---|
| Regression | $y = 0.75x + 0.01$ | $y = 0.87x + 0.04$ | $y = 0.63x + 0.01$ | $y = 0.84x + 0.05$ | $y = 0.85x + 0.05$ | $y = 0.83x + 0.04$ |
| r | 0.82 | 0.87 | 0.67 | 0.86 | 0.88 | 0.89 |
| F-statistic | 33.10 | 55.22 | 14.50 | 72.35 | 72.01 | 82.36 |
| *p*-value | $2.97 \times 10^{-5}$ (***) | $9.82 \times 10^{-7}$ (***) | 0.00129 (**) | $4.46 \times 10^{-8}$ (***) | $3.16 \times 10^{-8}$ (***) | $1.03 \times 10^{-8}$ (***) |
| RMSE | 0.05 | 0.04 | 0.05 | 0.05 | 0.04 | 0.03 |

** $p < 0.01$, *** $p < 0.001$.

The ETC-estimated STD errors of the six satellite-derived ET products are represented at the original spatial resolutions of these products (Figure 4). At this spatial distribution, none of the products achieved full accuracy in the whole region. Conspicuously, SSEBop shows a significant difference over space; that is, error estimation was very poor in the north but fairly good in the south. However, according to the statistics of the error estimates for all pixels, the median STD error of GLASS is the smallest at 0.10 mm/day; then, GLEAM is 0.18 mm/day; both MOD16 and ERA5 are at 0.19 mm/day. PML_V2 has higher errors of 0.24 mm/day; finally, the highest is SSEBop, which reaches 0.27 mm/day. To a certain extent, these findings are consistent with the conclusions reported in previous studies [5].

Moreover, the statistics of the standard deviation, correlation coefficient, and signal-to-noise ratio estimated by the ETC for each ET product (Figure 5) illustrate that GLASS performed best in all three indicators, while SSEBop is the worst. It is worth noting that the removal of the seasonal signal leads to a lower signal-to-noise ratio (Figure 5c); this is because the input of the ETC is an anomaly part relative to the mean value of the ET signal, which has a smaller dynamic range than the original ET value [47].

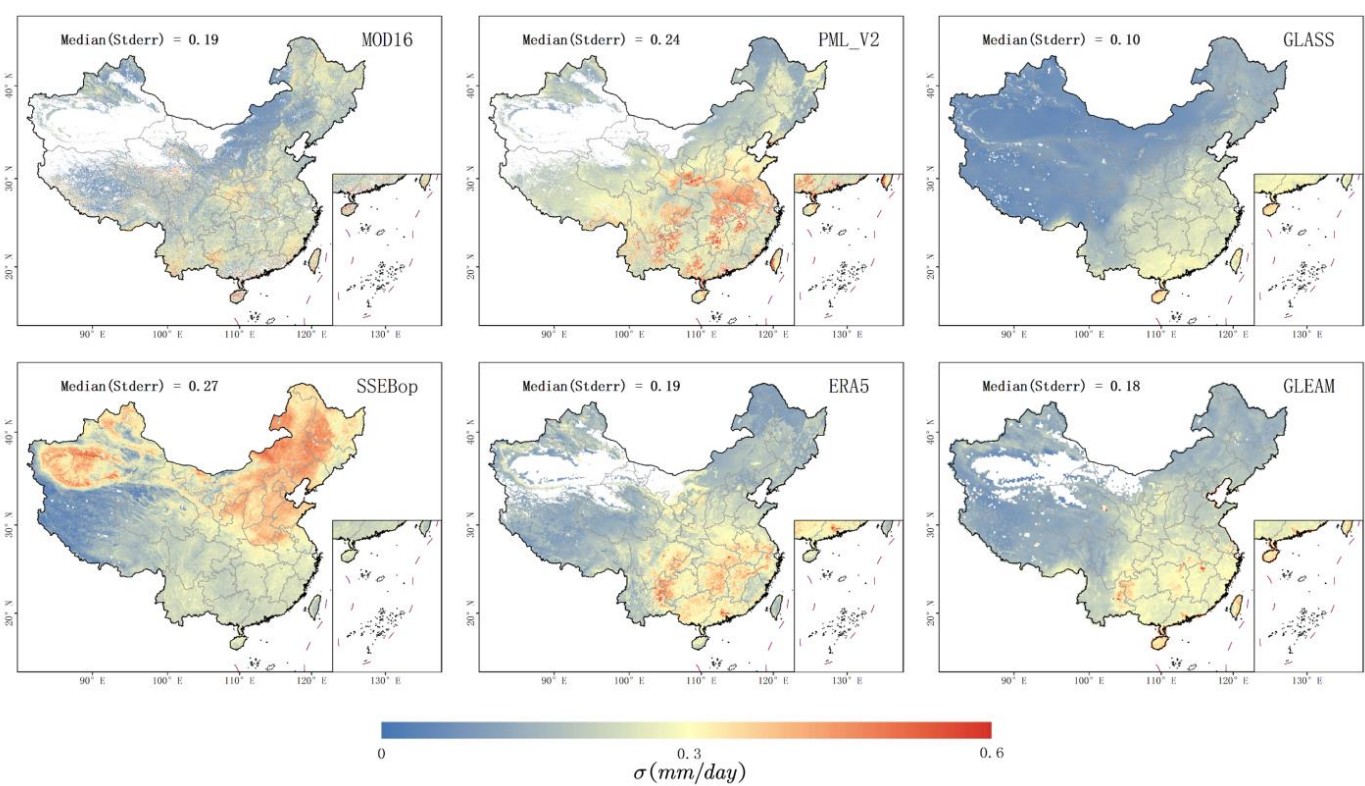

**Figure 4.** Spatial distribution of the ETC-estimated STD errors.

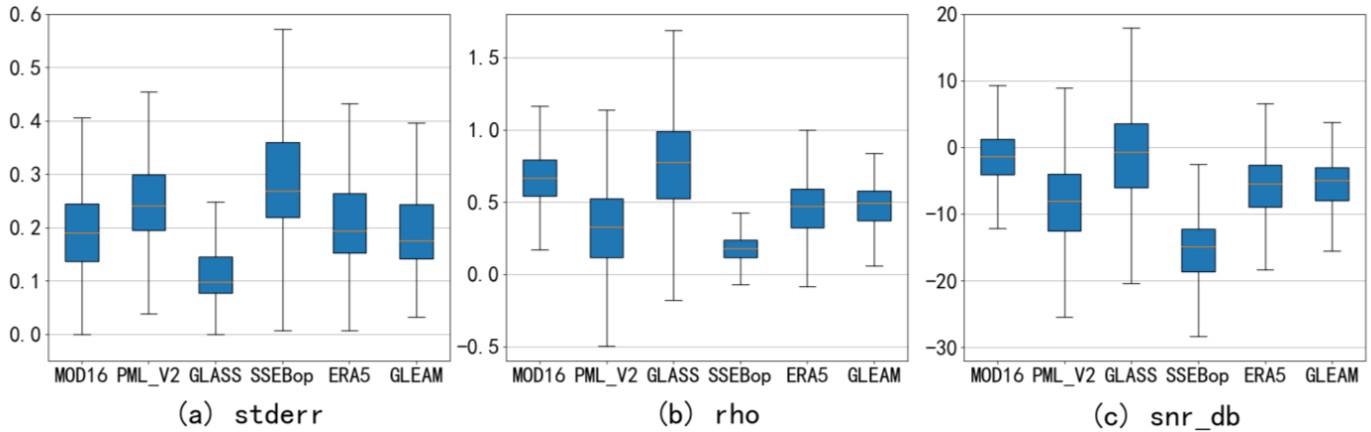

**Figure 5.** Boxplot of ETC statistics, (**a**) standard deviation of random errors, (**b**) the correlation coefficient, and (**c**) the signal-to-noise ratio.

### 4.3. Dataset Merging

Comparing the dominant weights of the six ET products on the pixels, the spatial pattern shows a relatively continuous patchy distribution (Figure 6). Among the products, GLASS performed the best, covering about 43.85% of the area; SSEBop took second place at about 35.24%, and the remaining four datasets accounted for no more than 10% each.

By utilizing the ETC-estimated errors, the ET products are integrated into a more precise dataset through a fuzzy membership weighting approach. It effectively captures the spatial heterogeneity across diverse regions, surpassing the capabilities of any individual ET product. The original ET data are merged at a resolution of 0.25° as an example according to the hardening and weighing methods, and the spatial distribution of two fusion outputs is highly consistent (Figure 7); however, the results obtained using the weighing approach are smoother or more continuous in space. The ET values of the merged dataset were

regressed against the measured ET values on 23 sites, finding that 20 sites were dominated by the weighing approach; as such, we recommend utilizing the weighing approach to merge data in China.

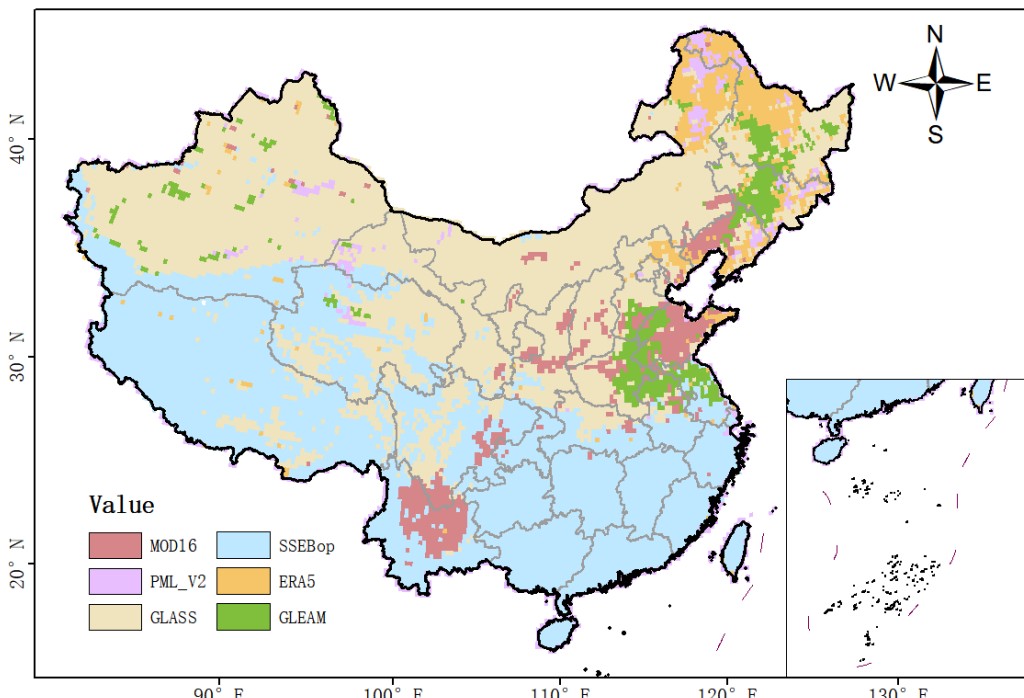

**Figure 6.** The dominant regions of the six ET products in the merged output.

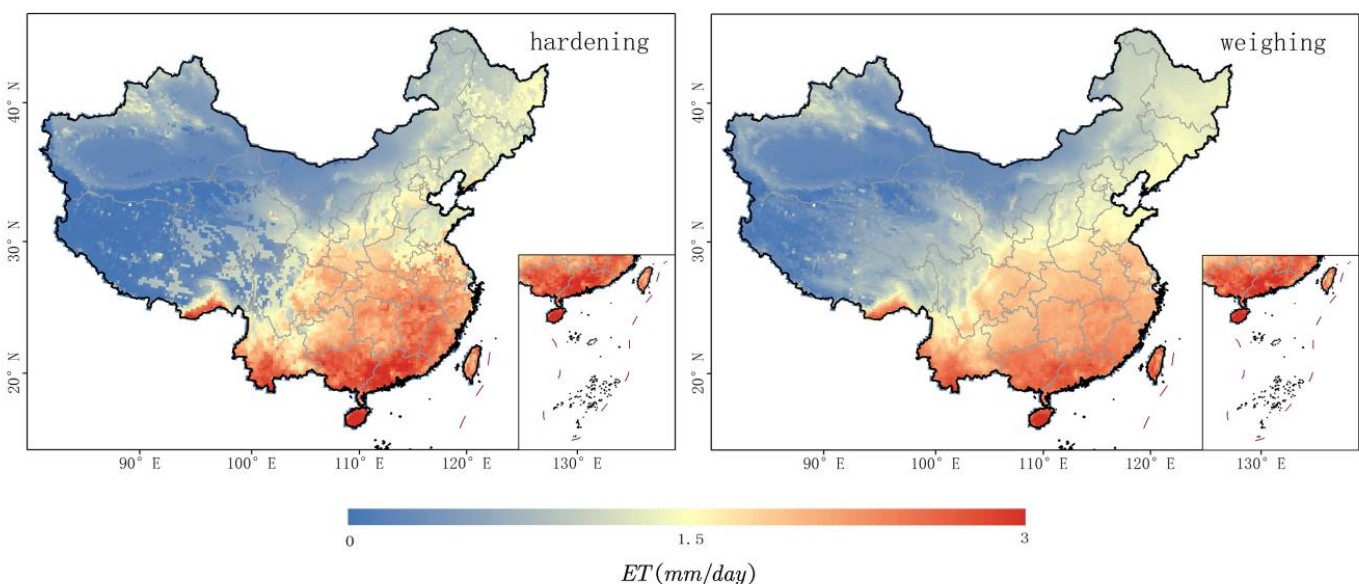

**Figure 7.** Two data fusion outputs.

The accuracy of the two fused ET products was assessed by comparing them with the six original datasets at 23 in-situ sites. The correlation between the weighted dataset and the measured values showed a significant improvement, with a median value of 0.79 across the in-situ sites, as shown in Figure 8.

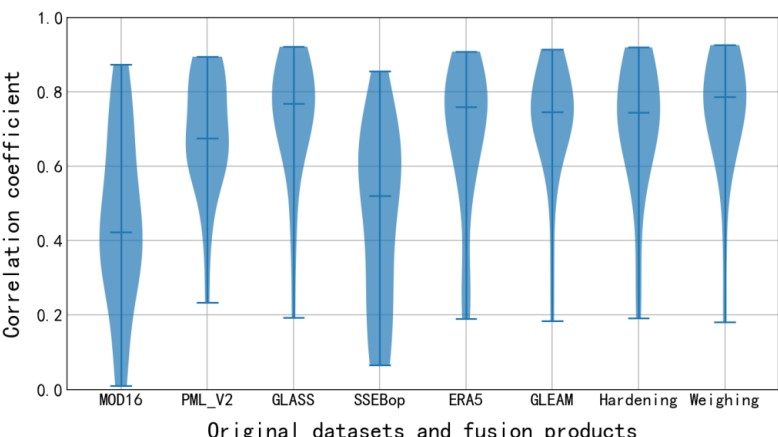

**Figure 8.** Correlation between ET datasets and measured values over 23 in-situ sites.

## 5. Discussion

### 5.1. The Impacts of Seasonal Changes on TC Analysis

Systematic errors (biases) caused by seasonal changes in ET signals usually affect the estimation of the STD of random errors in satellite-derived products. We compared two preprocessing approaches for lessening the impact of seasonal change on the TC analysis. The simpler method involves a fixed long-term mean signal [41], namely, removing a constant bias; however, this method cannot be used for signals with strong seasonal changes [40]. The other method involves selecting a seasonal change period as a filter window to remove the seasonal information of the dataset [42]. The correlation between the ETC-estimated STD errors and the actual STD errors at 23 in situ towers indicates that removing the seasonal bias leads to a higher correlation compared to the original dataset (Figure 9).

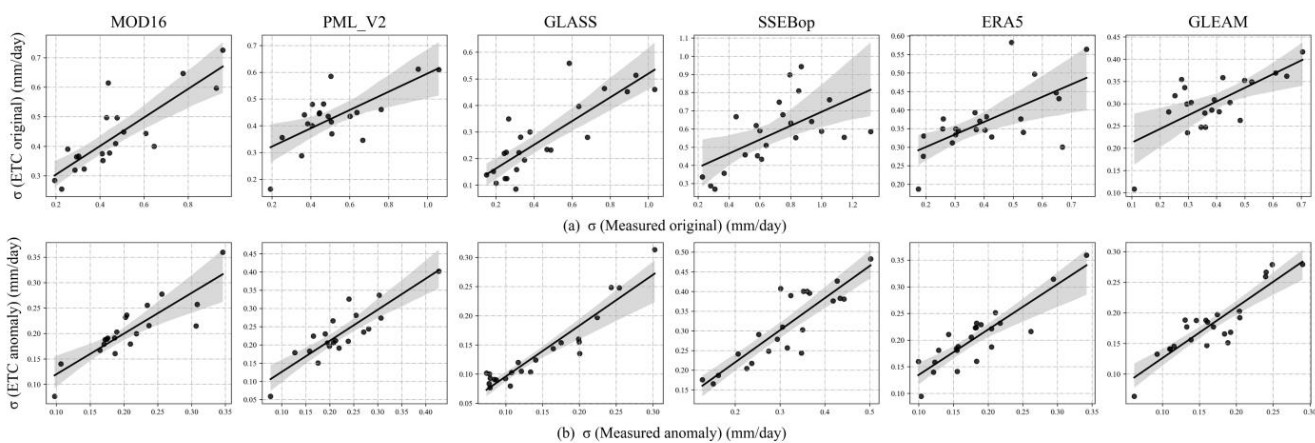

**Figure 9.** The impact of seasonal changes in ET on ETC-estimated STD errors: (**a**) Original dataset, and (**b**) Dataset with seasonal bias removed. The line and grey shadow of each scatter plot represent the best fit regression line and 95% confidence interval, respectively.

### 5.2. The Reliability of Triplet Ranking

The triplets of an ET product were ranked by comparing their performance at 23 in-situ sites; in this way, the best triplet for each product was determined. However, this comparison was conducted using limited observation points, and it remains necessary to determine whether the ranking of the triplets is valid for all pixels for each product.

A statistical approach to detecting the degree of the difference in the error estimates among the ranked triplets was proposed, in order to illustrate the reliability among the ranked triplets for all the pixels covered by the product. Assuming that higher-ranked error estimates are closer to actual errors than the lower-ranked ones, the difference between two higher-ranked errors must be less than that between a higher error and a lower error.

Therefore, we first selected three error estimates for each product, including the top two and the worst, then organized them into two groups—the "high–high" with two tops and the "high–low" with the worst and either of two tops. Finally, we calculated the absolute difference between the two group members.

As shown in Table 5, the means and STDs of the difference in the "high–high" groups are smaller than those in the "high–low" groups, except for the STD of MOD16. This finding reveals that the error estimates resulting from highly ranked triplets are more reliable than poorly ranked ones, across the entire region covered by the product. The results of the Kolmogorov–Smirnov Test also show that the difference between the two groups for each product is significant, as the *p*-value is far less than 0.05. Using the above way, we demonstrate that it is acceptable or practical to select the top ranked triplet as the optimum.

**Table 5.** The degree of the absolute difference in the error estimates among the ranked triplets.

| ET Products | "High–High" | | "High–Low" | | Kolmogorov–Smirnov Test | |
|---|---|---|---|---|---|---|
| | **Mean** | **STD** | **Mean** | **STD** | **Statistic** | **p-Value** |
| MOD16 (500 m) | 0.033 | 0.231 | 0.045 | 0.208 | 0.046 | 0.000 (***) |
| PML_V2 (500 m) | 0.007 | 0.122 | 0.022 | 0.147 | 0.079 | 0.000 (***) |
| GLASS (1 km) | 0.006 | 0.237 | 0.027 | 0.359 | 0.055 | 0.000 (***) |
| SSEBop (1 km) | 0.003 | 0.047 | 0.004 | 0.101 | 0.059 | 0.000 (***) |
| ERA5 (0.1°) | 0.003 | 0.046 | 0.046 | 0.199 | 0.137 | 0.000 (***) |
| GLEAM (0.25°) | 0.001 | 0.042 | 0.054 | 0.229 | 0.270 | 0.000 (***) |

*** $p < 0.001$.

In addition, the number of pixels for which the inputs in the triplet violate the TC assumption might, to some degree, indicate the quality of the triplets. As shown in Figure 10, the scatter plots show the violation of assumptions in the ETC analysis and the correlation coefficient between the TC-estimated and actual STDs at 23 observation sites. The plot indicates a certain degree of negative correlation; that is, the larger the number of violations, the more serious the triplet's violation of the TC assumption.

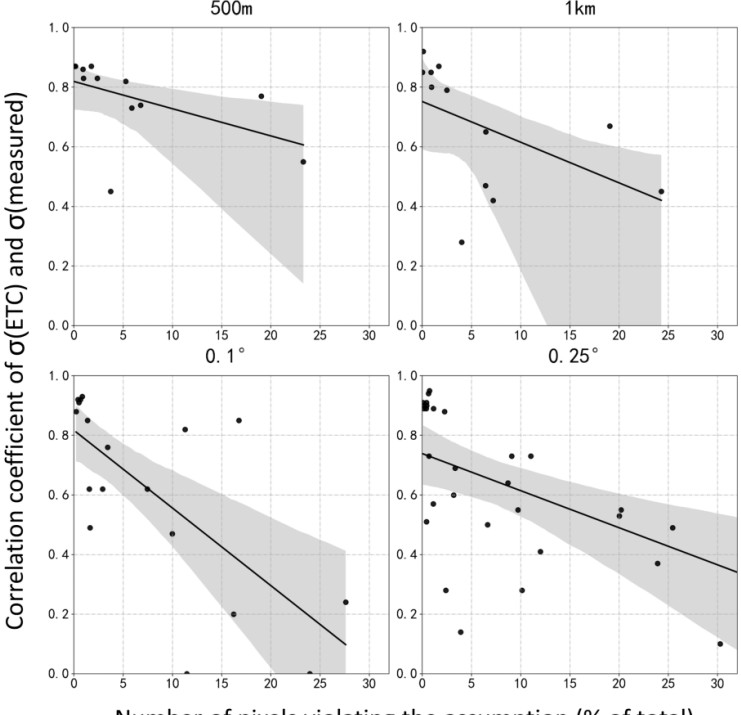

**Figure 10.** Scatter plots of TC assumption violations and the correlation coefficients between ETC-estimated STDs and actual STDs. The line and grey shadow of each scatter plot represent the best fit regression line and 95% confidence interval, respectively.

*5.3. The Issues of Identifying the Optimal Triplet*

The errors of remotely sensed or reanalyzed ET products are inevitably correlated as they often share ancillary datasets for their calculation. Nevertheless, the error estimation by ETC still achieved acceptable results on a global scale [3,15]. Regretfully, it is difficult to identify the best triplet or quadruplets under current advanced multiple collocation techniques [7–10], as there are no good ways to measure the non-zero error correlation among ET products in the absence of in-situ measures.

As suggested [12], a better error assessment can only happen when new, better, and independent data sources become available. In this experiment, the flux at 23 in-situ sites was acquired to support the error estimation. In most cases, the best-performing triples can be identified against all possible triplets, but there are exceptions. For the example of the 1 km SSEBop, the correlation between ETC-based and in-situ-based error metrics is 0.856 by the triplet MOD16-GLASS-SSEBop and 0.854 by PML_V2-GLASS-SSEBop, too close to determine which is better since the observations at these in-situ sites cannot represent the entire region, and the random error for a product in a grid-based dataset is also different to some extent to a point-based measurement [46].

The errors of the same ET product will change when converting from a very fine resolution to a quite coarse resolution; this is because the neutralization occurs on the aggregation of adjacent ET signals [11], which thus causes the spatial representativeness of the random errors for the same product differing in spatial resolution [46]. Such a difference caused by spatial aggregation can definitely affect the non-zero error correlation between two ET products, so this work sticks the optimal triplet tightly to a specific spatial resolution on which the ETC performed. However, the impact of spatial aggregation on the error correlation is difficult to capture and needs to be further explored.

## 6. Conclusions

Triple collocation (TC) is an effective method for quantifying data errors in ET products, but it is susceptible to the violation of its underlying assumptions, particularly arising from the input datasets. In this study, we identified the optimal triplet configuration that minimizes the violation of the non-zero cross-correlation hypothesis, improving the accuracy of error estimation. Performance ranking was conducted for all triplets comprising six widely used ET products, by correlating their ETC-estimated error metrics with in-situ error metrics at multiple flux observation points. The top-ranked triplet was determined as the optimum configuration, and the accuracy of this ranking was further validated through the statistical analysis of error estimate differences among the ranked triplets.

In the context of the experiment conducted in China, the error distribution of six datasets are very different in space. Generally, the GLASS product exhibited the lowest median error of 0.10 mm/day, indicating its superior performance. It was closely followed by the GLEAM, ERA5, and MOD16 products, which displayed median errors below 0.20 mm/day. The PML_V2 and SSEBop products demonstrated slightly higher median errors, measuring 0.24 mm/day and 0.27 mm/day, respectively. The weighted fusion ET product presents an accuracy improvement across all 23 in situ sites in comparison of the hardening way.

In the process of error assessment for ET products, our focus was on identifying the optimal triplets that minimize violations of the non-zero cross-correlation hypothesis through experimental investigation. However, the exact extent of the non-zero correlation remains uncertain. To achieve a more accurate assessment of errors, future research should incorporate new, improved, and independent data sources. Additionally, utilizing remote sensing datasets to explore the relationship between ET and different LULC categories is an intriguing way to consider in the future, as it indicates the impacts of vegetation growth on ET modeling and helps understand geographical errors.

**Author Contributions:** Conceptualization, Y.H., Z.D., R.L., M.W. and X.S.; Methodology, Y.H., H.M. and X.S.; Software, C.W. and J.H.; Validation, Y.H., C.W., J.H. and C.Q.; Formal analysis, Y.H., C.W., J.H. and X.S.; Investigation, Y.H., H.M., Z.D., R.L., M.W. and X.S.; Resources, H.M. and C.Q.; Data curation, Y.H. and X.S.; Writing—original draft, Y.H.; Writing—review & editing, X.S.; Visualization, Y.H. and C.W.; Supervision, X.S.; Project administration, C.Q., M.W. and X.S.; Funding acquisition, Z.D., R.L., M.W. and X.S. All authors have read and agreed to the published version of the manuscript.

**Funding:** This research was financially supported by the National Key Research and Development Program of China (No. 2020YFC1807103), the 973 Program (No. 2013CB733402), the National Natural Science Foundation of China (No. 42341206 and No. 40771167), Network Security and Informatization Special Application Demonstration Project of Chinese Academy of Sciences (No. CAS-WX2023SF-0403-01), and the Fundamental Research Funds for the Central Universities (No. E0E48914X2). In addition, Z.D. is grateful to have received funding from the Crafoord Foundation (No. 20200595 and No. 20210552).

**Data Availability Statement:** The source of data used in this study is provided in the manuscript.

**Acknowledgments:** We thank FLUXNET, the National Tibetan Plateau Science Data Center of China, and the National Ecological Science Data Center of China for providing in-situ measurements of evapotranspiration data, and the producers of the six evapotranspiration product datasets.

**Conflicts of Interest:** The authors declare no conflict of interest.

## Abbreviations

The following abbreviations are used in this manuscript:

| | |
|---|---|
| ET | Evapotranspiration |
| TC | Triple Collocation |
| ETC | Extended Triple Collocation |
| QC | Quadruple Collocation |
| MC | Multiple Collocation |
| STD | Standard Deviation |

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
