# Peer review of "Discovering Optimal Triplets for Assessing the Uncertainties of Satellite-Derived Evapotranspiration Products"

_remotesensing, doi:10.3390/rs15133215_

Round 1

Reviewer 1 Report (New Reviewer)

​​​​Dear authors,

I have carefully reviewed your article and while I found some of the findings to be promising, however, there are issues that need to be addressed. By addressing these issues, you can significantly enhance the scientific merit and impact of your study. I look forward to seeing your revised manuscript.

Its ok 

Author Response

Reviewer 2 Report (New Reviewer)

The article focuses on the evaluation and optimization of error estimations in evapotranspiration (ET) products, particularly satellite-derived ET products. The authors highlight the challenge of obtaining accurate error information due to limited validation data from flux towers. 

Overall, the data and the analysis are reliable, however, to my point of view, the article does not provide any solid conclusions, or give any assessment results of the evaporation product. The discussion and the conclusions primarily revolve around the feasibility of the TC method or the shortcomings of this method. The conclusions are not specific to the evaporation datasets in my opinion, and such similar conclusions can be used to evaluate other variables as well.

The authors aspect mentioned in the article is the use of the data fusion method to merge the ET products. However, the relevance and implications of this aspect are not clearly stated in the introduction, discussion, or conclusion sections of the article. It would be beneficial to provide more context and clarify the purpose of this work, highlighting how the data fusion method enhances the accuracy or utility of the ET products.

some minors:

L14 discovering to discover

L193-195 please rephrase the sentence 

L433 is the P<0.0001 really needed?

L466 to be further explored.

The English language is OK, some errors could be further revised.

Author Response

Reviewer 3 Report (New Reviewer)

Due to the difficulty in obtaining validation data, it is difficult to determine the error estimation of ET products based on satellite remote sensing. This study proposed a cross-correlation analysis approach to discovering the optimal triplet of satellite-derived ET products with regard to providing the most reliable error estimation. Based on this method, error estimates of six common ET products were calculated and compared with 23 observed data in situ flux towers. The paper is well organized and the results can support the conclusion. I recommend an acceptance.

Author Response

Dear reviewer,

Thank you for your positive feedback on our manuscript. We made significant efforts to ensure the reliability of the error estimation by identifying the optimal triplet of satellite-derived ET products. This time, we have thoroughly reviewed the manuscript and implemented several corrections to improve its overall quality and enhance its clarity.

Best regards,

Xianfeng Song , on behave of all coauthors

2023-06-18

Reviewer 4 Report (New Reviewer)

Comments on the “Discovering optimal triplets for assessing the uncertainties of satellite-derived evapotranspiration products” by Y. He et al.

The paper proposed a new approach, the cross-correlation analysis approach, to discover optimal triplets for assessing uncertainties of satellite-derived ET products. Validation of ET products is important, not only against in-situ observations but also using certain statistical methods. The subject of the paper is in line with the journal’s focus. The paper is well organized and written, and the results are well presented. I recommend accepting it for publication as long as the following minor issues are addressed.

1.     The authors selected 6 ET products. Why did the authors select those 6 ET products? What are the principles of selection? The study area is China, while the 6 ET products are all at a global scale. As far as I know, there are multiple ET products specially in China.

2.     L174-176: For the in-situ flux datasets, you have to cite which dataset and the corresponding paper, not just the data center. There are many datasets in each data center.

The English of the manuscript is good. 

Round 2

Reviewer 1 Report (New Reviewer)

 Accept in present form

No

This manuscript is a resubmission of an earlier submission. The following is a list of the peer review reports and author responses from that submission.

Round 1

Reviewer 1 Report

This manuscript tried triple collocation (TC) method to evaluate six ET products. I did not think TC is quite suitable in this context. Unlike precipitation and soil moisture products, ET products are generated based on multiple meteorological forcing and land surface datasets. These ancillary datasets can be shared by many products, violating the basic assumption of zero error correlations. For example, PML_V2 and SSEBop share the MODIS and GLDAS forcing datasets. Better to check it with quadruplets like one paper published in Remote Sensing of Environment. It is unreasonable to select the best triplets based on the correlation between ETC-based and in-situ-based error metrics. Please also check the numbers in the manuscript. The numbers are different in Figure 2 and the associated text. Numbers seem to be also questionable in Figure 7.

Reviewer 2 Report

 The paper submitted aims at assessing the uncertainty associated with the estimation of ET retrieved from different satellite products using a triplets of Triple Collocation methodology.

The paper is generally well structured considering the methodology which is well defined and described. My concerns are related to the discussion section which seems a pursuance of the results section. Indeed the paragraph is full of images and tables showing some results. I think that here the Authors need to strengthen the main strong points and weaknesses of the methodology proposed and support their statements with reference literature.

 Other comments:

- please add a glossary with all the abbreviation used in the text

- describe the presence of flux station in the area in paragraph 2.1

- figure 1 is not cited in paragraph 2.1. Moreover, some figures (e.g. in the discussion) are mentioned after their localization in the text. Please revise

- not all the equations are mentioned in the text

- lines 249-256 are not so clear because first is stated that the triplets are evaluated at the original spatial resolution, however, it is also stated that the GLASS was resampled

- in figure 2 MOD16 has a median of 0.19, while in the text is 0.24. Also in figure 2 GLASS has a median of 0.10, while in the text is 0.12

- please describe in the caption of figure 3 what a) b) and c) represent.